# Ecological Momentary Assessment of Awake Bruxism with a Smartphone Application Requires Prior Patient Instruction for Enhanced Terminology Comprehension: A Multi-Center Study

**DOI:** 10.3390/jcm11123444

**Published:** 2022-06-15

**Authors:** Laura Nykänen, Daniele Manfredini, Frank Lobbezoo, Antti Kämppi, Anna Colonna, Alessandra Zani, André Mariz Almeida, Alona Emodi-Perlman, Aslak Savolainen, Alessandro Bracci, Jari Ahlberg

**Affiliations:** 1Department of Oral and Maxillofacial Diseases, University of Helsinki, 00100 Helsinki, Finland; antti.kamppi@helsinki.fi (A.K.); jari.ahlberg@helsinki.fi (J.A.); 2School of Dentistry, Department of Biomedical Technologies, University of Siena, 53100 Siena, Italy; daniele.manfredini75@gmail.com; 3Department of Orofacial Pain and Dysfunction, Academic Centre for Dentistry Amsterdam (ACTA), University of Amsterdam and Vrije Universiteit Amsterdam, 1081 LA Amsterdam, The Netherlands; f.lobbezoo@acta.nl; 4Postgradute School of Orthodontics, University of Ferrara, 44121 Ferrara, Italy; annachiara.colonna@gmail.com; 5School of Dentistry, University of Padova, 35122 Padova, Italy; zani.alessandra@gmail.com (A.Z.); info@alessandrobracci.com (A.B.); 6Centro de Investigação Interdisciplinar Egas Moniz (CiiEM), Instituto Universitário Egas Moniz (IUEM), Quinta da Granja, Monte de Caparica, 2829-511 Caparica, Portugal; andremarizalmeida@gmail.com; 7Department of Oral Rehabilitation, The Maurice and Gabriela Goldschleger School of Dental Medicine, Tel-Aviv University, Tel-Aviv 6139001, Israel; dr.emodi@gmail.com; 8Department of Public Health, University of Helsinki, 00014 Helsinki, Finland; aslaks@welho.com

**Keywords:** awake bruxism, ecological momentary assessment, medical terminology comprehension

## Abstract

The prevalence of awake bruxism (AB) has been reported as being 30%, with sleep bruxism (SB) at 9–15%. Most studies have focused on SB, emphasizing the importance of AB research. For epidemiological evaluations of AB, a smartphone application based on ecological momentary assessment (EMA) was introduced. The aims of this multi-center study were: (1) to investigate how well lay subjects comprehend the AB terminology used in the smartphone application, and (2) to find out whether professional instruction improved their comprehension. The study population consisted of lay subjects from Italy, Portugal, and Finland comprising 307 individuals (156 men, 151 women; 18–86 years). Subjects first completed a five-item questionnaire about the meanings of the five AB terms used in the smartphone application. Each question offered four answer options, with one being correct. Immediately afterwards, the meanings of the terms were instructed. Lastly, the subjects were re-tested with the same questionnaire. In Finland and Italy, the re-tested correct answer scores for the single terms were at 89–97% per term. Improved comprehension was seen across sex, education, and age groups. In the Portuguese data, no improvement was found. Significant differences were found between countries in the improved scores for all terms that were correct following the instruction (Finland, 16.3% to 72.1%; Italy, 32.3% to 83.8%; Portugal, 23.1% to 33.7%) (*p* < 0.001). In conclusion, standardized instruction on AB terminology prior to EMA is recommended to improve the reliability of collected data.

## 1. Introduction

In a frequently cited consensus article, an international expert group defined bruxism as “a repetitive jaw-muscle activity characterized by clenching or grinding of the teeth and/or by bracing or thrusting of the mandible. Bruxism has two distinct circadian manifestations: it can occur during sleep (designated as sleep bruxism) or during wakefulness (designated as awake bruxism (AB)” [1]. In their recent update of the definition of bruxism, the expert group emphasized the involvement of masticatory muscle activities (MMA) during sleep and wakefulness [2]. MMA, if not associated with negative health outcomes, is considered harmless, while if it is associated with symptoms that negatively affect oral or general health, such as severe tooth wear or masticatory myalgia, it is considered harmful. MMA might also be associated with positive health outcomes, e.g., having a protective effect against harmful health conditions such as gastric esophageal reflux disease or obstructive sleep apnea, by stimulating the swallowing reflex or opening obstructed airways, respectively. However, the literature does not report any cut-off values for harmful levels of MMA, nor are such values hypothesized for the purported clinical consequences of bruxism [3].

According to the consensus definition, bruxism should not be considered as simply a rhythmic masticatory muscle activity linked to sleep arousal that associates with teeth grinding. Other bruxism behaviors, such as teeth clenching and mandible bracing, may also occur, and their non-negligible frequency during wakefulness offers a growing field for bruxism research. The changing bruxism paradigm has shifted to the concept of “muscle work”, which may present with a variable pattern that occasionally overloads the masticatory system more than periodic, nocturnal rhythmic masticatory muscle activity. The hypothesis of the new bruxism paradigm, as well as of this study, is that masticatory muscle activity during wakefulness may contribute to muscle-related symptoms of bruxism, such as fatigue, stiffness, and pain.

So far, AB has mainly been reported by patients or subjects using single-point self-report questionnaires in both clinical and research settings. Across Western populations, the prevalence of AB has been reported to be up to 30%, with SB reported in 9–15% [4]. The higher prevalence, and potentially harmful effects, of AB on oral and general health indicate the importance of learning more about it. For the purpose of epidemiological evaluations, a smartphone application (BruxApp^©^) [5] was introduced to take advantage of so-called ecological momentary assessment (EMA) [6]. Such an approach is not a novelty in the field of psychological sciences, and has gained popularity in medical science as well [7,8,9]. This smartphone application for AB allows for the real-time collection of self-report data on the momentary presence of five different oral conditions. The first is “relaxed jaw muscles” (a normal condition) and the other four, “mandible bracing”, “teeth contact”, “teeth clenching”, and “teeth grinding”, are typical AB behaviors. The reporting period is usually one week. At present, the application gives an alarm sound up to twenty times during wakefulness, which prompts the subject to tap on their momentary oral condition immediately. Based on compliance studies undertaken by the developers, at least twelve alarms per day must be responded to in order to consider the daily data collection valid. The application has been translated into 26 languages, and several large-scale studies are in preparation. Preliminary studies examining the feasibility of the application with small population samples have produced promising results [10,11,12,13]. One recent large-scale study successfully used the application to collect data on AB [14]. The reliability and validity of the application as a scientific and clinical tool is currently being assessed. A basic pre-requisite is that AB terminology is understood by users, which has yet to be evaluated in a lay population. Better comprehension of the terminology means more accurate results when using the application. Considering that, in the literature, AB’s role in causing bruxism-related symptoms has remained unknown, studies using EMA are needed to gather more valid data on AB. This could enhance clinicians’ work when treating bruxism-related symptoms.

Based on these premises, the aims of this study were: (1) to use a questionnaire to test lay subjects’ comprehension of AB terms used in the smartphone application to describe five oral conditions related to AB behaviors, and (2) to investigate whether giving instruction in AB terminology after the first questionnaire would provide improved understanding in a second, identical questionnaire completed immediately after the instruction. The study hypotheses were (1) that the lay subjects’ comprehension of AB terminology is poor, and (2) that comprehension would significantly improve after instruction.

## 2. Materials and Methods

The study was conducted in Finland, Italy, and Portugal. The English version of BruxApp^©^ [4], a smartphone application (see Introduction), was translated into Finnish and Portuguese for the study, using the World Health Organization protocol: “Process of translation and adaptation of instruments” [15]. The Italian version of the application already existed.

In each country, the study consisted of three parts, beginning with the administration of a five-item questionnaire. Each item was a question regarding the meaning of an AB term used in the smartphone application and had four response options, one of which was correct. The first AB term described a normal oral condition, while the other four terms described awake masticatory muscle activities typical of AB. Without prior instruction, subjects had to choose the answer that they thought was correct. In the second part of the study, which followed immediately, subjects received a brief lesson on AB terminology, including the correct definition of each AB term. This took place face-to-face with a clinician (Italy), or using a pre-recorded video created by a trained clinician (Finland and Portugal). The English version of the video can be seen here: https://youtu.be/wOjnXHLzj0E, accessed on 1 May 2020). For the third and final part of the study, the subjects completed a questionnaire identical to the first one. An e-questionnaire (Microsoft Forms^©^ https://forms.office.com/Pages/ResponsePage.aspx?id=WXWumNwQiEKOLkWT5i_j7lOreBq5YX5GgUvsRhiyA11UNEU4TjA5TFdVMDU2NU9UUzU3T1FaN1oxVS4u, accessed on 1 May 2020) was used in Finland and Portugal.

In Italy, the study was conducted by dental professionals, either one-to-one or in small group sessions in dental practice waiting rooms, where the paper version of the questionnaire was administered, and face-to-face instruction was performed. In all countries, data collection took place in the autumn of 2020, within a time frame of three months.

The terms used in both questionnaires were the same as in the EMA smartphone application BruxApp^©^, describing the five oral conditions the subject had to choose between when responding to an alarm. The terms were: (I) relaxed jaw muscles, (II) mandible bracing, (III) teeth contact, (IV) teeth clenching, and (V) teeth grinding. The response options are shown in Table 1. It should be noted that when using the e-questionnaire, it was not possible for the subjects to discuss or change their initial answers; neither was it possible to submit the form unless it was completed.

The aim of the study was to gather at least one hundred replies from each country. In Finland, professional musicians (n = 104; non-patients) in three symphony orchestras from the Uusimaa province, southern Finland, participated as part of a follow-up study on sleep-related issues. Subjects were invited by email or phone. In Italy, patients (n = 99) were recruited from two private dental practices in Padova and Siena. Exclusion criteria were a previous diagnosis of temporomandibular disorder (TMD) or an age under 18 years. In Portugal, the participants (n = 104) were patients referred to a hospital tertiary clinic (ear–nose–throat). Exclusion criteria were a previous diagnosis of TMD or an age under 18 years.

The study was reviewed and granted permission by the Ethical Research Committee of the Medical Faculty of the University of Helsinki (permit no 01/2020).

### Statistical Methods

Statistical analyses were performed with the Statistical Package for Social Sciences (Version 25, SPSS Inc., Chicago, IL, USA). The Wilcoxon signed ranks test was used to compare the changes between the answers before and after instruction. Each data item was categorized and computed as follows: (test: correct answer = 1, incorrect = 0), and all items pooled as: (test: all correct = 1, incorrect = 0; re-test: all correct = 1, incorrect = 0).

## 3. Results

Of the 307 participants, 151 were women and 156 were men, with ages ranging from 18 to 86 years. The percentage of answers to each question by country is shown in Table 1, and by sex, age, and education in Table 2.

Subjects in all three countries had some awareness of AB terminology prior to instruction. In Finland and Italy, the instruction session appeared to improve results irrespective of the method used (i.e., face–to–face, or pre–recorded video). The post-instruction comprehension for single terms was positive at slightly over 90% per item, with little variation based on sex, age group, or educational level. No such improvement in comprehension was found in the Portuguese data.

In Finland and Italy, the comprehension of each individual term (I–V) significantly improved after the instruction session (*p* values, I: <0.001, II: <0.001, III: 0.005, IV: 0.001, V: <0.001). The flow chart is shown in Figure 1.

Moreover, significant differences (*p* < 0.001) were found in each country in the improved test–re-test scores for having all terms correct (Finland: 16.3% to 72.1%; Italy: 32.3% to 83.8%; Portugal: 23.1% to 33.7%), irrespective of sex or age group. When the data from the three countries were pooled, the corresponding test–re-test scores improved from 23.8% to 62.9% (*p* < 0.001) (Figure 1).

## 4. Discussion

The present study showed that the comprehension of individual AB terms among the study subjects improved following a brief instruction session. The score for answering all terms correctly also markedly improved.

With a reported prevalence up to 30% in adult populations, AB has usually been assessed by subjective single-point self-reports. Certain specific clinical signs in the oral cavity may also imply the presence of this behavior (e.g., indentation linguae, linea alba on buccal mucosa, etc.) To gather data on AB, especially considering the possible time-variant nature of bruxism behaviors, patients may be asked to keep a diary for 1–2 weeks to record their masticatory muscle activity (viz., teeth clenching, mandible bracing/thrusting, etc.) [2]. However, with both research and clinical settings in mind, a smartphone application to track AB behaviors has recently been introduced. This is based on so-called ecological momentary assessment [5], an approach that relies on receiving repeated random alerts during wakefulness. Thus far, the app appears promising for the study of real-time AB behaviors and their correlates on a large scale [10,11,12,13,14], but further studies are needed to assess its reliability and validity. A Standardized Tool for the Assessment of Bruxism (STAB) is in preparation, which could assist in the validation protocol of the smartphone application in future studies [16].

Two pilot studies to test the smartphone application’s general functioning principles with small samples of healthy, young Italian adults described the frequency of various oral conditions related to AB. The findings suggested that EMA may be a viable tool for evaluating AB behaviors [10,12]. However, despite the promising potential of smartphone-based EMA strategies, it should be noted that they are based on lay persons’ subjective self-reports, and knowing how to convey accurate information from the professional to the patient and vice versa is essential.

As the reliability and validity of the application have not been fully tested before, the present study examined subjects’ comprehension of the AB terminology used. Although instruction significantly improved comprehension at the group level, some variation between the countries was observed. While the majority of subjects correctly replied after the instruction session, there were some who went from a correct to an incorrect answer. This is rather difficult to interpret, and requires further attention in future validation studies.

It may be that the subjects’ comprehension improved due to increased familiarity with the terms as a result of the instruction, or simply due to repetition of the terms. To distinguish between these mechanisms will require studying subjects with no previous instruction, i.e., performing the test–re-test questionnaires without the instructional aspect. This might reveal that the repetition of terms was a significant factor in the improved re-test results. A similar setup could be used to test the impact of instruction alone, by ruling out any effect of repetition. However, notwithstanding a possible repetition effect, it should be borne in mind that in practice, both in research and clinical environments, the smartphone application is always used as per the instructions provided.

At present, the smartphone application has been translated into 26 languages, using the English version as the source. The first published translation was in Polish [7], providing a road map for translations into other languages. The translations from English into Finnish and Polish were completed according to a protocol of the World Health Organization: “Process of translation and adaptation of instruments” [15]. This protocol emphasizes the cross-cultural and conceptual, rather than linguistic/literal, equivalence of the translation.

Studies have shown that lay subjects in general tend to have a low comprehension of medical terminology [17,18,19]. In addition, it may be easier for a lay subject to identify a behavior than to understand the concept behind it. Studies employing a method similar to the present study (modified test–re-test with teaching of the terminology by video) have been found useful; for example, comprehension prior to instruction was unambiguously poor in the field of prostate cancer and chemotherapy terminology, but improved after it [20,21]. It has also been shown that lower educational levels correlate with poorer comprehension of common prostate terms [22]. This suggests the importance of taking into account the educational level of subjects in this kind of investigation. This study also observed that subjects with higher educational levels had better comprehension at the group level than others, both pre- and post-instruction. However, in the present study, the impact of educational level should be interpreted with caution because of its dichotomous and hence ambiguous categorization.

The subjects in the present study were very heterogeneous, and not to be considered representative from nationwide or global perspectives. As well as being different nationalities, the subjects were professional symphony orchestra musicians, dental patients, or hospital patients at a tertiary clinic. Despite this, the findings indicated quite similar trends in improved comprehension after instruction. A potential explanation for the somewhat different results from the Portuguese data is that they may be due to the complexity of the Portuguese language, which often uses the same expression for several meanings. However, differences between the Portuguese, Finnish, and Italian results cannot be explained by the instructional modality, viz., face-to-face rather than using the video. Nevertheless, the pre-recorded video may have influenced the results; although it was the same for all subject’s content-wise, the videos in Portugal and Finland were made by different professionals and possibly at a different educational level. Other social and cultural factors might also be involved that were not considered as variables in this study’s design. Although such factors are not easy to control, at least comparing educational levels between study cohorts should be considered, among other variables.

The strength of our study was its multi-center approach. Furthermore, the method used in our study would be repeatable in other similar studies. A limitation was that we could not find a plausible explanation for the somewhat poorer comprehension of the AB terms in Portuguese subjects. It is advisable to control such issues in future studies.

It is worth emphasizing that when trying to obtain accurate and reliable data with the smartphone application (BruxApp^©^), whether for scientific or clinical purposes, it is vital to ensure that study subjects or patients comprehend the terminology of the different AB behaviors that professionals use to label them. If not, scientific interpretations risk being inaccurate and might produce hard-to-interpret or biased data. In clinical settings, bruxism assessment might be misinterpreted, which could lead to mistakes in diagnostics, and thus mismanagement of patients.

## 5. Conclusions

In conclusion, the present study shows that instruction prior to performing EMA is useful and can be given either face-to-face or by pre-recorded video. It would therefore be advisable to embed a standardized instructional video in the smartphone application that is mandatory to watch prior to the recording period, and that can be re-watched at any time. This would ensure more reliable results in future large-scale studies, and potentially make clinical assessment of AB with EMA more accurate.

## Figures and Tables

**Figure 1 jcm-11-03444-f001:**
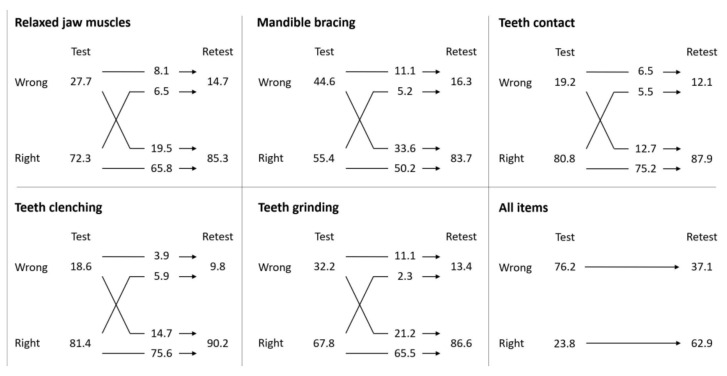
Summarized answers to the questions per item and to all items (percentages).

**Table 1 jcm-11-03444-t001:** Test-retest answers to the five studied items in Finland, Italy, and Portugal. Numbers of replies.

	Finland		Italy		Portugal	
n = 307	n = 104		n = 99		n = 104	
	Test	Retest	Test	Retest	Test	Retest
**Relaxed jaw muscles**						
(a) The mouth is fully open and you are not using any strength with the masticatory muscles	2	1	9	2	4	3
(b) The upper and lower teeth are slightly touching each other, without any relevant strength in the masticatory muscles	13	5	6	2	14	21
(c) The upper and lower teeth are kept apart for 1-4mm, independent of the masticatory muscles contraction	7	3	18	0	12	8
**(d) The teeth are slightly kept apart and the masticatory muscles are relaxed**	**82**	**95**	**66**	**95**	**74**	**72**
**Mandible bracing**						
(a) The mouth can be opened and closed with difficulty	39	0	7	0	15	9
**(b) The jaw muscles are kept contracted, with the mandible in a fixed position and without teeth contact**	**46**	**97**	**62**	**92**	**62**	**68**
(c) The lips are touching each other without teeth contact	9	4	12	4	11	8
(d) The movements of the mouth can be performed from one side to another without moving the teeth	10	3	18	3	16	19
**Teeth contact**						
(a) The upper and lower teeth are firmly kept together with the maximum force from the masticatory muscles	2	1	6	1	5	6
**(b) The teeth are slightly touching, even episodically, without any object or food between the two arches**	**93**	**96**	**82**	**95**	**73**	**79**
(c) The teeth are kept together with something between the upper and lower arch	3	1	6	3	4	4
(d) The teeth are kept together for more than five seconds	6	6	5	0	22	15
**Teeth clenching**						
(a) The mouth is opening and closing with repeated movements with maximum force from the masticatory muscles	0	2	6	0	3	3
(b) The teeth are touched with minimum force exerted by the masticatory muscles	22	4	8	2	8	10
**(c) The teeth are kept together firmly for some seconds or more**	**80**	**97**	**81**	**96**	**89**	**84**
(d) The mouth’s opening and closing movements are limited during wakefulness because the teeth were clenching the night before	2	1	4	1	4	7
**Teeth grinding**						
(a) An individual has the habit of moving repeatedly the jaw from one side to another without teeth contact	3	1	2	3	1	0
**(b) An individual moves repeatedly the jaw from one side to another and/or forwards and backwards, with teeth contact**	**47**	**89**	**82**	**92**	**79**	**85**
(c) A patient with pain in the face makes sounds with the teeth only during the night	11	4	6	1	8	4
(d) An individual moves repeatedly the jaw with such a teeth contact that is always audible	43	10	9	3	16	15

Correct answers and corresponding figures marked in **bold**.

**Table 2 jcm-11-03444-t002:** Test-retest answers by gender, age, and level of education. Percentages.

	Gender				Age						Education			
	Men		Women		18–30		31–45		>45		Academic		Other	
n = 307	n = 156		n = 151		n = 41		n = 144		n = 122		n = 250		n = 57	
	Test	Retest	Test	Retest	Test	Retest	Test	Retest	Test	Retest	Test	Retest	Test	Retest
**Relaxed jaw muscles**														
(a)	5.8	2.6	4.0	1.3	4.9	0.0	6.9	4.2	2.5	0.0	3.2	0.8	12.3	7.0
(b)	14.1	10.3	7.3	7.9	7.3	7.3	13.9	10.4	8.2	8.2	10.0	7.2	14.0	17.5
(c)	12.2	5.1	11.9	2.0	17.1	7.3	9.0	2.1	13.9	4.1	11.2	3.6	15.8	3.5
**(d)**	**67.9**	**82.1**	**76.8**	**88.7**	**70.7**	**85.4**	**70.1**	**83.3**	**75.4**	**87.7**	**75.6**	**88.4**	**57.9**	**71.9**
**Mandible bracing**														
(a)	19.9	3.8	19.9	2.0	14.6	2.4	17.4	4.2	24.6	1.6	20.8	3.2	15.8	1.8
**(b)**	**51.9**	**83.3**	**58.9**	**84.1**	**78.0**	**87.8**	**52.8**	**80.6**	**50.8**	**86.1**	**56.8**	**85.6**	**49.1**	**75.4**
(c)	14.1	6.4	6.6	4.0	0.0	4.9	10.4	5.6	13.9	4.9	8.8	4.4	17.5	8.8
(d)	14.1	6.4	14.6	9.9	7.3	4.9	19.4	9.7	10.7	7.4	13.6	6.8	17.5	14.0
**Teeth contact**														
(a)	3.8	5.1	4.6	0.0	9.8	4.9	2.8	3.5	4.1	0.8	3.2	2.4	8.8	3.5
**(b)**	**79.5**	**83.3**	**82.1**	**92.7**	**85.4**	**85.4**	**79.9**	**88.2**	**80.3**	**88.5**	**84.0**	**88.4**	**66.7**	**86.0**
(c)	5.1	2.6	3.3	2.6	0.0	0.0	6.3	2.1	3.3	4.1	2.8	1.2	10.5	8.8
(d)	11.5	9.0	9.9	4.6	4.9	9.8	11.1	6.3	12.3	6.6	10.0	8.0	14.0	1.8
**Teeth clenching**														
(a)	3.2	3.2	2.6	0.0	7.3	4.9	0.7	2.1	4.1	0.0	1.6	1.6	8.8	1.8
(b)	10.9	7.1	13.9	3.3	12.2	7.3	11.1	6.3	13.9	3.3	11.6	4.0	15.8	10.5
**(c)**	**80.8**	**86.5**	**82.1**	**94.0**	**78.0**	**82.9**	**84.0**	**88.2**	**79.5**	**95.1**	**84.4**	**92.4**	**68.4**	**80.7**
(d)	5.1	3.2	1.3	2.6	2.4	4.9	4.2	3.5	2.5	1.6	2.4	2.0	7.0	7.0
**Teeth grinding**														
(a)	3.8	1.3	0.0	1.3	0.0	2.4	2.1	0.7	2.5	1.6	2.0	0.8	1.8	3.5
**(b)**	**67.3**	**85.3**	**68.2**	**88.1**	**87.8**	**90.2**	**75.7**	**87.5**	**51.6**	**84.4**	**69.2**	**89.6**	**61.4**	**73.7**
(c)	7.1	1.9	9.3	4.0	4.9	7.3	6.3	2.1	11.5	2.5	7.2	3.2	12.3	1.8
(d)	21.8	11.5	22.5	6.6	7.3	0.0	16.0	9.7	34.4	11.5	21.6	6.4	24.6	21.1

Correct answers and corresponding figures are in **bold**. Answer alternatives a–d equal with Table 1.

## Data Availability

All data can be obtained from the corresponding author.

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
