# Peer review of "Ecological Momentary Assessment of Awake Bruxism with a Smartphone Application Requires Prior Patient Instruction for Enhanced Terminology Comprehension: A Multi-Center Study"

_jcm, 2022, doi:10.3390/jcm11123444_

Round 1

Reviewer 1 Report

Ecological Momentary Assessment of Awake Bruxism with a 2 Smartphone Application Requires Prior Patient Instruction for 3 Enhanced Terminology Comprehension: A Multi-Center Study

The paper is well written, with very good english, with na importante theme nowadays that applicatives are contributing to health treatment.

The introduction is well written, explanation over parafunction and the smartphone application are were very detailed.

The aims of the study were well defined.

Methodolgy: the terms used in the applicative were not proper to lay patients, they are difficult for someone not in the health field to undestand.

Results: In the results the answer of the patients were correct and else answer. What was considered correct answer? How do they know it was the correct answer?

Was there any clinician to respond to any questions of the patients regarding the app in Finland and Portugal? Was the attention given to patients better in Italy? This information is missing.

There is a missing number of protocol approval: Ethical Research Committee of the Medical Faculty of the University of Helsinki

Also, the difference between ages of patients among the three countries were not shown in any of the tables.

In table 1 there is a high number of patients having relaxed muscles as the same time a high number of patients having contracted jaw muscles. It is a contradiction, please explain.

There is a high variantion between the answers between the countries. Finland: 16.3% to 72.1%; Italy:32.3% 162 to 83.8%; Portugal:23.1% to 33.7%). This high variation is not adequately discussed why between the countries.

Also I would recomend to include the variables face-to-face and video results to show that there were not a statistical difference between them, because it could be a viés of the study.

Author Response

The paper is well written, with very good english, with na importante theme nowadays that applicatives are contributing to health treatment.

The introduction is well written, explanation over parafunction and the smartphone application are were very detailed.

The aims of the study were well defined.

We wish to thank the reviewer for the positive comments.

Methodolgy: the terms used in the applicative were not proper to lay patients, they are difficult for someone not in the health field to undestand.

Terms used in the application are common language (i.e., teeth clenching), not medical terms (i.e., tonic bruxism). We did hypothesize that lay people may have difficulties comprehending them, thus we aimed to find out whether giving instructions would improve the comprehension.

Results: In the results the answer of the patients were correct and else answer. What was considered correct answer? How do they know it was the correct answer?

Out of the four answer alternatives, one was correct, and others were incorrect (” else”). We changed “else” into “incorrect” throughout the text. Originally, “else” refers to a SPSS term. The application does not give feedback whether the answer is correct or not. The application is designed to collect data on subjects’ oral status. Our study aimed to improve the quality of the application in data gathering.

Was there any clinician to respond to any questions of the patients regarding the app in Finland and Portugal? Was the attention given to patients better in Italy? This information is missing.

Thank you for this very good point. Study subjects in Italy had a possibility to ask questions from the training clinician during the face-to-face education. Subjects in Finland and Portugal did not, as they watched a video. This is not visible in the results, though, as Italy and Finland had similar results and Portugal much poorer. Please find this in results, paragraph 1.

There is a missing number of protocol approval: Ethical Research Committee of the Medical Faculty of the University of Helsinki

We added it to the manuscript, thank you for the notice.

Also, the difference between ages of patients among the three countries were not shown in any of the tables.

We did not consider it necessary, as the analyses showed age had no effect on comprehension in the pooled data. Please see Table 2.

In table 1 there is a high number of patients having relaxed muscles as the same time a high number of patients having contracted jaw muscles. It is a contradiction, please explain.

We have now clarified the text in Methods, paragraph 2. Study subjects were asked the meaning of the terms, not their actual oral status. Meaning of all five terms were asked separately.

There is a high variation between the answers between the countries. Finland: 16.3% to 72.1%; Italy:32.3% 162 to 83.8%; Portugal:23.1% to 33.7%). This high variation is not adequately discussed why between the countries.

We have tried to interpret the poorer results in Portugal, but unfortunately, found no plausible explanation apart from the possible poorer educational quality of the Portuguese video. Please note that age, educational level, and gender variation was similar in all three countries.

Also I would recomend to include the variables face-to-face and video results to show that there were not a statistical difference between them, because it could be a viés of the study.

We carefully studied this comment and found that the reply already is in Results, page 5, first paragraph.

Reviewer 2 Report

The paper needs revisions as per the following recommendations before publication.

Abstract: 

1. revise the aims or objective of the study....it should start with an action verb...to determine, demonstrate, evaluate, compare, summarize etc.

2. The conclusion part needs revision, mention the main findings of the study as per objective in your own words...avoid statistical statements or differences or p-value, etc.

3. throughout the manuscript...please write the probability value in italic p-value.

4. Mention hypothesis information in the introduction section only...where it is applicable.

Introduction:

1. please elaborate what is the need of doing this research? how this study is going to help clinicians, dentists, or doctors. mention the rationale for doing this study in the last paragraph.

Discussion:

mention the limitation, strengths, and future recommendations of the study in the last paragraph.

Conclusions:

The main conclusion is missing in the paper

References:

the references need revisions as per journal format.

Author Response

We wish to thank the reviewer for the constructive comments and have revised the manuscript according to the reviewers’ suggestions. Please note in the text that the language has been edited by a native speaking language editor specialized in medical writing. 

We also carefully went through the references and consider them relevant.

Abstract: 

  1. revise the aims or objective of the study....it should start with an action verb...to determine, demonstrate, evaluate, compare, summarize etc.

We have corrected this in the manuscript.

  1. The conclusion part needs revision, mention the main findings of the study as per objective in your own words...avoid statistical statements or differences or p-value, etc.

We have now separated the conclusion from the results in the abstract, in the conclusion part we present no statistical statements or p-values.

throughout the manuscript...please write the probability value in italic p-value.

Thank you for noting this, we have corrected it throughout the manuscript.

  1. Mention hypothesis information in the introduction section only...where it is applicable.

We removed the hypotheses from the abstract and elaborated them more clearly in reference to need of doing this study (please see answer to next comment).

Introduction:

  1. please elaborate what is the need of doing this research? how this study is going to help clinicians, dentists, or doctors. mention the rationale for doing this study in the last paragraph.

We have added a paragraph to the Introduction as per the reviewer’s recommendation.

Discussion:

mention the limitation, strengths, and future recommendations of the study in the last paragraph.

We have added a paragraph to the discussion mentioning the limitations, strengths, and future recommendations, as per suggested by the reviewer.

Conclusions:

The main conclusion is missing in the paper

We have now separated our conclusions into its own paragraph in Discussion.

References:

the references need revisions as per journal format.

We have corrected the references as per the journal format. We apologize for the mishap.

Reviewer 3 Report

Fine multicentre research

It seems that computer diagnostics develops medicine in many programs which allows to train the system of starting medical interviews.

At work, I miss the level of patient education in chapter materials. How does it affect the understanding of the AB problem.

An objection may be added to other reports of AB frequency in member states.

I would add an introductory note to the title due to the need to conduct research and the system -preliminary report

It is for this type of high level of marked patients, even those who are not affected by AB.

Author Response

We wish to thank the reviewer for the overall positive comments. Below are our point-to-point answers.

It seems that computer diagnostics develops medicine in many programs which allows to train the system of starting medical interviews.

At work, I miss the level of patient education in chapter materials. How does it affect the understanding of the AB problem.

We assume the reviewer is referring to the educational level of study subjects. The comprehension was indeed poorer in non-academic subjects both before and after instruction, which is supported by another study (ref 22). This can be found in the Discussion.

If the reviewer refers to the instruction material, the video (in English) can be seen in https://youtu.be/wOjnXHLzj0E, as it is also mentioned in the manuscript.

An objection may be added to other reports of AB frequency in member states.

There is very limited, if any data on AB frequency in general populations. Preliminary studies have been done with BruxApp, that are referred to in the manuscript (refs no 10, 12, 14). BruxApp is designed to serve AB epidemiology research on a global level.

I would add an introductory note to the title due to the need to conduct research and the system -preliminary report

It is for this type of high level of marked patients, even those who are not affected by AB.

We thank for this suggestion but unfortunately, we cannot consider this as a preliminary study on AB epidemiology nor behaviors, as we tested the comprehension of AB terminology in lay subjects. We see it as preliminary study to future large-scale studies using Brux-App. We hope this reasoning can be accepted by the referee.

Round 2

Reviewer 2 Report

The authors have improved the paper, I would suggest to write conclusion under separate heading....no 5 !!

This manuscript is a resubmission of an earlier submission. The following is a list of the peer review reports and author responses from that submission.

Round 1

Reviewer 1 Report

Title :A multi-center study on comprehension of awake bruxism ter- 2 minology: prior instructions are advisable for a successful Eco- 3 logical Momentary Assessment by a smartphone application

1)Abstract: it must well structured :You have to specify the purpose materials and methods and conclusion and the purpose is not clear

2)introduction : the purpose is not clear

3)  Materials and Methods are not clear it should be rewritten for the type of study too small the sample

4)the conclusions are missing

Author Response

Reply: We thank the reviewer for the valuable work done to improve the quality of our work. In part due to the review report form, it is somewhat difficult to figure out how to response to the ‘must be improved’ comments. However, of course, we carefully studied all comments made by the referee and did our best to clarify the text whenever noticed anything to correct. Please take note, that the reviewer 2 did not demand similar modifications. But for example, the language was rechecked by a native speaking British professional (shown in the Acknowledgements). However, in our reply we focus on the detailed points given by the reviewer. Please find our point-by-point response below, as well as the revised manuscript in a separate file.

1)Abstract: it must well structured :You have to specify the purpose Materials and Methods and conclusion and the purpose is not clear

Reply: We have clarified the purpose and the conclusion. We consider that the materials and methods are ok.

2)introduction : the purpose is not clear

Reply: Please see above.

3)  Materials and Methods are not clear it should be rewritten for the type of study too small the sample

Reply: With all respect, we do not fully understand this comment. Material and methods, after a careful check, look ok in our opinion. Neither should the sample size be a problem.

4)the conclusions are missing

Reply: We clarified the conclusion both in the abstract and in the text.

Reviewer 2 Report

Dear Authors,

the aim of your research could be interesting, however, some weaknesses in the methods deserve to be dealt with.

You administered the video/explanation and IMMEDIATELY tested the results. did you think about the administration of the second questionnaire after, for example, few months? is it possible that laymen immediately understand but not learn?

What about the different methods of administration of contents? this issue may affect the soundness of results. Furthermore, bruxism is an important disease and it deserves to be well explained to patients and they have to be well aware of the pathology. So, a stronger approach to the teaching is advisable

Author Response

Reply: We thank the reviewer for the valuable comments to improve the quality of our work. Overall, the tick box comments in the review report form are a bit difficult to work on. Thus, we focused on the detailed points made by the reviewer. Please see our point-by-point reply below, as well as our revised manuscript. However, we are grateful that the reviewer considered the Introduction pertinent. We carefully checked all other general points made and improved the text whenever considered possible. Conclusions should now be more clearly written.

The aim of your research could be interesting; however, some weaknesses in the methods deserve to be dealt with.

Reply: It is difficult to reply to this unspecified comment, viz., ‘some weaknesses’. We had a careful look at the text, nevertheless. However, we do not see any need to rewrite the section.

You administered the video/explanation and IMMEDIATELY tested the results. did you think about the administration of the second questionnaire after, for example, few months? is it possible that laymen immediately understand but not learn?

Reply: The reviewer has a good point in this. However, in the present study we tested a feasible in-built education method (vs. face-to-face instructions) of the app. In the field test studies, the app is to be used immediately after the test-retest protocol for seven subsequent days. All this is targeted to improve understanding of awake bruxism behaviors. The results in our study showed that this act independently improved the responses. This can also be important performing EMA in large populations. In future studies however, numbers under preparation, it is necessary to take into account all what the reviewer points out.

What about the different methods of administration of contents? this issue may affect the soundness of results. Furthermore, bruxism is an important disease and it deserves to be well explained to patients and they have to be well aware of the pathology. So, a stronger approach to the teaching is advisable

Reply: Indeed, there is a lot of work ahead before we know more of the changing paradigm of bruxism, the etiology and consequences. Its harmless-risk-protective facets should be borne in mind, both in research and clinical work. Therefore, to add a piece in the awake bruxism puzzle, to better understand involuntary masticatory muscle activities, the BruxApp was developed.

Round 2

Reviewer 1 Report

Compared to the previous version it is not changed much :

Title: it is too long and it is not understandable 

Abstract : it is not well structured , it needs to be rewritten and made understand the purpose and conclusions 

materials and methods : the sample number is too small 

Conclusion : should be expanded